# Comparison of the L3-23K and L5-Fiber Regions for Arming the Oncolytic Adenovirus Ad5-Delta-24-RGD with Reporter and Therapeutic Transgenes

**DOI:** 10.3390/ijms26083700

**Published:** 2025-04-14

**Authors:** Aleksei A. Stepanenko, Anastasiia O. Sosnovtseva, Marat P. Valikhov, Anastasiia A. Vasiukova, Olga V. Abramova, Anastasiia V. Lipatova, Gaukhar M. Yusubalieva, Vladimir P. Chekhonin

**Affiliations:** 1Department of Fundamental and Applied Neurobiology, V. P. Serbsky National Medical Research Center of Psychiatry and Narcology, Ministry of Health of the Russian Federation, 119034 Moscow, Russia; sollomia@yandex.ru (A.O.S.); marat.valikhov@gmail.com (M.P.V.); nastjavasukova@gmail.com (A.A.V.); abramova1128@gmail.com (O.V.A.); chekhoninnew@yandex.ru (V.P.C.); 2Department of Medical Nanobiotechnology, Institute of Translational Medicine, N. I. Pirogov Russian National Research Medical University, Ministry of Health of the Russian Federation, 117997 Moscow, Russia; 3Center for Precision Genome Editing and Genetic Technologies for Biomedicine, Engelhardt Institute of Molecular Biology, Russian Academy of Sciences, 119991 Moscow, Russia; lipatovaanv@gmail.com; 4Laboratory of Regulation of Intracellular Proteolysis, Engelhardt Institute of Molecular Biology, Russian Academy of Sciences, 119991 Moscow, Russia; 5Laboratory of Cell Technologies, Federal Scientific and Clinical Center for Specialized Types of Medical Care and Medical Technologies, Federal Medical and Biological Agency of the Russian Federation, 115682 Moscow, Russia; kakonya@gmail.com; 6Laboratory of Immunotherapy of Solid Tumors, Federal Center for Brain and Neurotechnology, Federal Medical and Biological Agency of the Russian Federation, 117513 Moscow, Russia; 7Laboratory of Molecular Mechanisms of Regeneration, Engelhardt Institute of Molecular Biology, Russian Academy of Sciences, 119991 Moscow, Russia

**Keywords:** delta-24-RGD, glioblastoma, glioma, hyaluronidase PH20, immunotherapy, L3-23K, L5-Fiber, oncolytic adenovirus 5, soluble PD-1 ectodomain, suppressor of RNA silencing p19

## Abstract

The insertion of a transgene downstream of the L3-23K or L5-Fiber region was reported as a vital strategy for arming E3 non-deleted oncolytic adenoviruses. However, depending on the percentage of codons with G/C at the third base position (GC3%) and the type of splicing acceptor, an insert downstream of the L5-Fiber region may substantially affect virus fitness. Since the insertion of transgenes downstream of the L3-23K and L5-Fiber regions has never been compared in terms of their expression levels and impact on virus fitness, we assessed the total virus yield, cytolytic efficacy, and plaque size of Ad5-delta-24-RGD (Ad5Δ24RGD) armed with *EGFP*, *FLuc*, the suppressor of RNA silencing *p19*, soluble wild-type human/mouse and high-affinity human programmed cell death receptor-1 (PD-1/*PDCD1*) ectodomains, and soluble human hyaluronidase PH20/*SPAM1*. The insertion of transgenes downstream of the L3-23K region ensures their production at considerably higher levels. However, the insertion of transgenes downstream of either region differentially and unpredictably affects the oncolytic potency of Ad5Δ24RGD, which cannot be explained by GC3% or expression level alone. Surprisingly, the expression of the human and mouse PD-1 ectodomains with 83.1% and 70.1% GC3%, respectively, does not affect cytolytic efficacy but increases the plaque size in a cell line-dependent manner.

## 1. Introduction

Among all viruses adapted for antitumor therapy, oncolytic adenoviruses (OAds), which are based on the human adenovirus 5 genome, occupy a dominant position in preclinical research and clinical trials [1,2]. A number of OAds with improved specificity of replication (cell type-dependent tight transcriptional control) [3], infection efficacy (e.g., insertion of a targeted peptide in the fiber or fiber chimerism) [4,5], and the kinetics of cell death and progeny release (e.g., knockout of *E1B-19K*, C-terminal truncation mutations in the *i-leader* or *E3-19K*, or overexpression of the adenovirus death protein-encoding gene *ADP*) [6,7] have been designed. Moreover, a wide variety of therapeutic transgene-armed OAds with improved in vivo efficacy have been reported [2,8,9,10,11], and several armed OAds have entered clinical trials [2,8].

The packaging capacity of adenoviral capsids is restricted. Because only ≈2 kb of extragenomic DNA (≈105% of the size of the wild-type genome) can be efficiently and stably incorporated into Ad5 capsids [12], the deletion of individual adenovirus genes that do not significantly affect replication and the total progeny yield is necessary to insert transgene(s) with regulatory sequences of a total larger size. The partial or nearly complete deletion of early transcription unit 3 (E3), encoding largely proteins with immune–inhibitory and immune evasion functions, has become the main strategy for constructing armed OAds. However, OAds with partial deletions of the E3 region were reported to be cleared much more rapidly in immunocompetent mouse and hamster tumor models [13,14].

Another much less commonly exploited approach for arming OAds is to place a transgene under the control of endogenous transcriptional regulatory sequences of the adenovirus genome instead of using bulky autonomous transgene cassettes with exogenous promoters and poly(A) signal sequences. Splicing acceptors (SAs) have proven themselves sufficient for this purpose, given their small size and the provision of high transgene production under the regulation of the major late promoter (MLP), which is responsible for the control of transcription of the major late transcription units (L1–L5). According to Farrera-Sal et al. [15], four different SAs were exploited to express transgenes from OAds: an SA from the Ad5 protein IIIa gene (IIIaSA), an SA from the Ad40 or Ad41 long fiber gene (40SA or 41SA), and an artificial SA derived from the beta globulin gene (BPSA). In the E3 region non-deleted OAds, two positions for inserting a transgene were tested: downstream of the 23K protease gene (L3-23K region) [16] and downstream of the fiber gene (L5-Fiber region) [15,17,18]. To our knowledge, these two positions have never been compared with each other for arming OAds in terms of their impact on virus fitness and the level of transgene production.

A higher percentage of codons with G or C at the third base position (GC3%) of transgenes inserted downstream of the L5-Fiber region was shown to be associated with more impaired oncolytic potency [15,19]. Specifically, the enhanced green fluorescent protein (*EGFP*) and click beetle green luciferase reporter transgenes with 96.7% and 57.1% GC3% inserted downstream of the L5-Fiber region of the wild-type Ad5 or ICOVIR-15K genome, respectively, negatively affected the cytolytic efficacy in A549 cells [15,19]. ICOVIR-15K is an E1AΔ24-based OAd with four palindromic E2F binding sites and one SP1 binding site in the E1A promoter and an RGDK motif replacing the putative KKTK heparan sulfate glycosaminoglycan-binding domain in the fiber shaft. Moreover, codon deoptimization of *EGFP* from a GC3% of 96.7% to 44.8% or ≈60% or, conversely, codon optimization of the bee hyaluronidase and bispecific T-cell engager (BiTE) against fibroblast activation protein (FAP) transgenes resulted in an improvement or deterioration, respectively, of virus fitness in A549 cells [19]. The authors suggested that the design of OAds armed with a therapeutic transgene under the control of the MLP should include the optimization of transgene expression and virus fitness through balanced transgene codon usage (GC3%) to reduce transgene–viral intergenic competition and preserve viral lytic activity [15,19].

In addition, the authors compared splice acceptors IIIaSA and 40SA for arming ICOVIR-15K with the click beetle green luciferase, the FAP-targeting BiTE, and a soluble version of the human hyaluronidase posterior head 20 (hPH20/*SPAM1*) downstream of the L5-Fiber region [15]. On the one hand, inserting coding sequences of these transgenes with 40SA induced higher production levels than when IIIaSA was used [15]. On the other hand, the cytolytic efficacy of ICOVIR-15K armed with the luciferase or FAP-targeting BiTE was better with IIIaSA than with 40SA in A549 cells (≈six-fold and ≈forty-fold, respectively, lower IC_50_ values), whereas no substantial differences in IC_50_ values (less than two-fold) were observed between IIIaSA and 40SA in the *hPH20*-armed ICOVIR-15K in A549 cells and the five additional cell lines tested [15]. Since the GC3% for luciferase, FAP-targeting BiTE, and hPH20 were 57.1%, 59.2%, and 35.6%, respectively, the authors concluded that the luciferase and FAP-targeting BiTE might compete more strongly with late viral genes for cellular translational resources than the hyaluronidase hPH20 does, resulting in more pronounced interference with virus fitness, especially when 40SA is used [15]. 40SA was proposed as a tool to enhance the production of transgenes with relatively low GC3% content [15], since lower GC3% content of the coding sequence affects messenger RNA (mRNA) stability and translation efficiency [20]. Finally, the insertion of any transgene affected the cytolytic potency. Compared with the parental virus, ICOVIR-15K armed with IIIaSA and luciferase, FAP-targeting BiTE, or *hPH20* presented 140-, 5-, and 3.8-fold higher IC_50_ values, respectively, in A549 cells [15]. Unfortunately, this study did not compare the total virus yields and plaque phenotypes (the sizes of vital dye-stained plaques) of armed OAds.

Since all these findings were suggested for use as a design guide for arming OAds [15,19], we performed a comparative analysis of the clinically advanced Ad5-delta-24-RGD armed with five reporter and therapeutic transgenes using 40SA or IIIaSA (depending on GC3% of a transgene) with insertion sites downstream of the L3-23K or L5-Fiber region. Our data indicate that transgene insertion downstream of the L3-23K region results in much higher levels of transgene production than transgene insertion downstream of the L5-Fiber region. However, transgene insertion downstream of both the L3-23K and L5-Fiber regions has differential and unpredictable effects on virus fitness (e.g., total virus yield, cytolytic efficacy, and plaque size), which cannot be fully interpreted based on only the GC3% and transgene production levels.

## 2. Results

### 2.1. Generation of Transgene-Armed Oncolytic Adenoviruses

We previously reported the generation de novo [21] of Ad5-delta-24-RGD (or Ad5∆24RGD herein), which is an E1AΔ24-based [22,23] infectivity-enhanced E3 region non-deleted OAd with the insertion of the arginine–glycine–aspartic acid (RGD) motif-containing integrin-targeting peptide RGD-4C in the HI loop of the fiber knob domain [24] (Figure 1A). Ad5∆24RGD (DNX-2401, tasadenoturev) has been tested in phase I/II clinical trials in patients with high-grade glioma and presented a favorable toxicity profile [25,26,27,28]. We used Ad5∆24RGD as a platform to incorporate five transgenes: *EGFP* (total of 239 amino acid residues, aa), firefly luciferase (*Fluc*, total of 550 aa), the suppressor of RNA silencing *p19* of tomato bushy stunt virus (Tombusvirus genus) with a C-terminal FLAG tag (*p19*FLAG, total of 180 aa), the soluble ectodomain of human programmed cell death receptor-1 (hPD-1/*PDCD1*, 26–147 aa) with the native signal peptide (1–25 aa) replaced by the murine immunoglobulin kappa (IgGκ) light chain signal peptide (total of 142 aa), and the soluble C-terminal truncated (20 aa) hPH20/*SPAM1* (total of 489 aa). All the transgenes were inserted into a cassette containing 40SA (for *p19*FLAG and *hPH20*; GC3%, 38.3%, and 35.8%, respectively) or IIIaSA (for *EGFP*, *Fluc*, and IgGκ-ecto-hPD1; GC3%, 97.1%, 49.6%, and 83.1%, respectively), a Kozak consensus sequence, and a poly(A) signal sequence overlapping with a stop codon (Appendix A) downstream of the L3-23K [16] and L5-Fiber [15] regions at sites corresponding to nucleotides 22,354 (Figure 1B) and 32,798 (Figure 1C), respectively, of the wild-type Ad5 genome.

### 2.2. The Insertion of Transgenes Downstream of the L3-23K Region Ensures Their Production at Considerably Higher Levels

We first quantitatively compared the activity of the EGFP and Fluc transgenes inserted downstream of the L3-23K and L5-Fiber regions. A flow cytometry analysis of human lung adenocarcinoma A549 and glioblastoma LN18 cells at 24 h postinfection (hpi) and murine glioma GL261 and CT-2A cells at 48 hpi with serial dilutions of Ad5Δ24RGD_IIIa-EGFP (L3-23K) and Ad5Δ24RGD_IIIa-EGFP (L5-Fiber) revealed that the median fluorescence intensity (MFI) of EGFP expressed downstream of the L3-23K region was on average 1.6–1.8-fold higher in A549 cells, 2.6–3.2-fold higher in LN18 cells, 6.7–9.1-fold higher in GL261 cells, and 5.0–5.8-fold higher in CT-2A cells (Figure 2A). We confirmed these observations by analyzing the luminescence intensity (relative light units [RLUs]) of cells infected with serial dilutions of Ad5Δ24RGD_IIIa-Fluc (L3-23K) and Ad5Δ24RGD_IIIa-Fluc (L5-Fiber) (Figure 2B). The RLU for Fluc expressed downstream of the L3-23K region was on average 3.7–15.3-fold higher at 24 hpi and 2.3–5.2-fold higher at 48 hpi in A549 cells, 4.3–14.5-fold higher at 24 hpi and 1.9–6.5-fold higher at 48 hpi in LN18 cells, 4.3-fold higher at 24 hpi and 19–91-fold higher at 48 hpi in GL261 cells, and 5–47-fold higher at 24 hpi and 17–44-fold higher at 48 hpi in CT-2A cells (Figure 2B). Thus, although both the L3-23K and L5-Fiber regions are under the control of the MLP, and IIIaSA was used to express both reporter transgenes, the activity of EGFP and Fluc inserted downstream of the L3-23K region was considerably higher than that of EGFP and Fluc inserted downstream of the L5-Fiber region in both human and mouse tumor cells.

Next, we analyzed the in vivo activity of Fluc inserted downstream of the L3-23K region using a bioluminescence imaging system (IVIS). The CT-2A cells were preinfected with Ad5Δ24RGD_IIIa-Fluc (L3-23K) and stereotaxically inoculated into the brains of syngeneic mice. The bioluminescence intensity (radiance) reached a plateau on Day 2 postinjection, did not change considerably on Day 3, and was ≈10-fold higher than that on Day 1 postinjection and ≈100-fold higher than that of the background, with complete signal attenuation by Day 6 (Figure 2C and Appendix A). However, we observed more heterogeneous and less intense bioluminescence signals upon a single intratumoral injection of Ad5Δ24RGD_IIIa-Fluc (L3-23K) on Day 14 after the intracerebral inoculation of CT-2A glioma cells (Figure 2D and Appendix A). Nevertheless, a weak signal was detected in four out of five mice even on Day 6 after the virus injection (Appendix A). Thus, the activity of the Fluc reporter inserted downstream of the L3-23K region was preserved both when preinfected murine CT-2A glioma cells were placed in the normal brain tissue microenvironment and in the complex tumor microenvironment upon the intratumor injection of OAd.

Further, we compared the level of hPD-1 ectodomain secretion using an enzyme-linked immunosorbent assay (ELISA) and hPH20 hyaluronidase activity using a turbidimetric assay in the supernatants of infected A549 cells. We detected 63- and 68-fold more ecto-hPD-1 at 48 and 72 hpi with Ad5Δ24RGD_IIIa-IgGκ-hPD1 (L3-23K) than with Ad5Δ24RGD_IIIa-IgGκ-hPD1 (L5-Fiber) (Appendix A). No hPD-1 ectodomain was detected in the supernatant of cells infected with the parental virus Ad5Δ24RGD. Moreover, we readily detected hyaluronidase hPH20 activity in the supernatant of cells infected with Ad5Δ24RGD_40SA-hPH20 (L3-23K) after 45 min of incubation with hyaluronic acid at room temperature; however, hyaluronidase activity was not detectable for Ad5Δ24RGD_40SA-hPH20 (L5-Fiber) (Appendix A) and the parental virus Ad5Δ24RGD. We then infected 30-fold more cells with Ad5Δ24RGD_40SA-hPH20 (L5-Fiber), collected the supernatant at 48 hpi, and concentrated it 15-fold. The incubation with hyaluronic acid for 45 min again failed to detect hPH20 activity, but activity was confirmed after 16 h of incubation at room temperature (Appendix A). Finally, we confirmed p19FLAG expression in the lysates of cells infected with Ad5Δ24RGD_40SA-p19FLAG (L3-23K) and Ad5Δ24RGD_40SA-p19FLAG (L5-Fiber) by a Western blot analysis with antibodies against the FLAG tag (Appendix A). Taken together, the insertion of different transgenes downstream of the L3-23K region ensures their production at substantially higher levels than does insertion downstream of the L5-Fiber region.

### 2.3. The Insertion of Transgenes Downstream of the L3-23K or L5-Fiber Region Differentially and Unpredictably Affects the Oncolytic Potency of Ad5Δ24RGD

We performed a comparative analysis of the oncolytic potency of five pairs of transgene-armed OAds compared with that of the parental virus Ad5∆24RGD. The total yield of all OAds with transgenes inserted downstream of the L3-23K region, except for the hPH20-armed OAd, was considerably lower (on average 3.1–20-fold in A549 cells and 2.7–39-fold in LN18 cells, depending on the transgene), with the most negative effect being exerted by the insertion of IIIa-Fluc and 40SA-p19FLAG in both cell lines (Figure 3A). In contrast, the total yields of OAds with transgenes inserted downstream of the L5-Fiber region did not differ substantially from those of the parental virus (Figure 3A).

The cytolytic efficacy, as assessed by the fold change in the mean IC_50_ values obtained from survival curves of cells infected with serial dilutions of OAds, was reduced by an average of 3.7–4.0-fold for OAds with IIIa-EGFP (L3-23K), IIIa-EGFP (L5-Fiber), and IIIa-Fluc (L5-Fiber) insertions in A549 cells and 2.4-fold for OAd with IIIa-EGFP (L3-23K) insertion in LN18 cells (Figure 3B). Interestingly, the cytolytic efficacy was increased by an average of 3.5–4.2-fold and 2.2–3.2-fold for OAds with 40SA-p19FLAG (L5-Fiber) and 40SA-hPH20 (L5-Fiber) insertions in A549 and LN18 cells, respectively. For all other armed OAds, the difference was ≤two-fold that of the parental virus (Figure 3B).

The mean sizes of plaques stained with the vital dye thiazolyl blue tetrazolium bromide (MTT) were considerably reduced for OAds with 40SA-p19FLAG (L3-23K) and 40SA-hPH20 (L3-23K) insertions but only in LN18 cells (Figure 3C). Interestingly, the mean sizes of plaques were considerably increased for OAds with IIIa-IgGκ-ecto-hPD1 (L3-23K), IIIa-IgGκ-ecto-hPD1 (L5-Fiber), and 40SA-hPH20 (L5-Fiber) insertions but only in A549 cells (Figure 3C). OAds with IIIa-EGFP and IIIa-Fluc insertions did not form MTT-stained plaques that could be accurately quantified by Day 6 in A549 cells or by Day 12 in LN18 cells. In contrast, previously generated Ad5Δ24RGD_E1B55K-p2A-EGFP and Ad5Δ24RGD_DBP-p2A-EGFP [21] with the EGFP coding sequence fused to the C-terminus of E1B-55K or DBP via a p2A peptide formed MTT-stained plaques of similar mean size to the parental virus (Appendix A). An analysis of the size of the fluorescent plaques revealed that the spread of Ad5Δ24RGD_E1B55K-p2A-EGFP and Ad5Δ24RGD_DBP-p2A-EGFP was considerably more efficacious than that of Ad5Δ24RGD_IIIa-EGFP (L3-23K) and Ad5Δ24RGD_IIIa-EGFP (L5-Fiber), with Ad5Δ24RGD_IIIa-EGFP (L3-23K) exhibiting the lowest spread efficacy (Appendix A). Finally, the mean sizes of the MTT-stained plaques formed by OAds with IIIa-EGFP and IIIa-Fluc insertions downstream of the L3-23K region were considerably smaller than those formed by OAds with insertions downstream of the L5-Fiber region in A549 cells on Day 10 (Appendix A).

Overall, we can summarize as follows. (1) In contrast to transgene insertion downstream of the L5-Fiber region, insertion downstream of the L3-23K region affected the total infectious progeny yields in most cases. (2) A reduction in the total yield did not lead to a decrease in the cytolytic efficacy in most cases. (3) The cytolytic efficacy or the total yield of OAds in general did not associate with the efficacy of plaque formation and the size of plaques. (4) Regardless of insertion downstream of the L3-23K or L5-Fiber region, IIIa-EGFP and IIIa-Fluc substantially affected virus fitness. (5) The insertion of 40SA-p19FLAG only downstream of the L5-Fiber region improved the cytolytic efficacy but did not impact the total virus yield. (6) The insertion of 40SA-hPH20 downstream of the L5-Fiber region did not affect the virus fitness; however, the relatively low level of hyaluronidase activity in the supernatant should be considered. (7) The insertion of IIIa-IgGκ-ecto-hPD1 with a GC3% of 83.1% of the coding sequence downstream of the L3-23K or L5-Fiber region did not affect the cytolytic efficacy but surprisingly increased the size of plaques in a cell line-dependent manner. The relationship between GC3% content and amino acid length of the transgenes inserted downstream of the L3-23K and L5-Fiber regions and viral fitness indicators (total virus yield, cytolytic activity, and plaque size) are graphically summarized in Appendix A.

### 2.4. Characterization of Ad5Δ24RGD Expressing the Wild-Type Murine PD-1, Human PD-1 or the Human Mutant High-Affinity PD-1 Ectodomain

Since Ad5∆24RGD_IIIa-IgGκ-hPD1 generally displayed superior oncolytic potency among OAds with transgene insertions downstream of the L3-23K region, we analyzed OAds producing PD-1 ectodomains in more detail. We additionally obtained OAds with the insertion of the murine PD-1 ectodomain (25–147 aa) with the IgGκ signal peptide (IIIa-IgGκ-ecto-mPD-1, total 144 aa, 70.1% GC3%), as well as with the insertion of the human mutant high-affinity PD-1 ectodomain (designated here as hPD-1_HA_, with HA indicating high affinity) with the IgGκ signal peptide (IIIa-IgGκ-ecto-hPD-1_HA_, total 142 aa, 83.1% GC3%) (Appendix A). Peptides [29] and high-affinity mutants of the hPD-1 ectodomain [30] are considered alternatives to antibodies that target hPD-L1 as competitive antagonists. One of the high-affinity hPD-1 ectodomains with 10 amino acid substitutions generated by directed evolution was reported to bind hPD-L1 with a K_D_ value of 107 pM, as measured by surface plasmon resonance, in contrast to the wild-type hPD-1 ectodomain with a K_D_ value of 3.88 µM [30]. We generated Ad5Δ24RGD_IIIa-IgGκ-hPD1_HA_ (L3-23K) with the insertion of this high-affinity hPD-1 ectodomain. We confirmed the secretion of the hPD-1 and hPD-1_HA_ ectodomains in the supernatants of infected human and murine cells, with the latter being substantially less efficient producers (Figure 4A). For example, compared with A549 cells, murine glioma GL261 and CT-2A cells produced 11- and 17-fold lower levels of the hPD-1 ectodomain, respectively. We also confirmed that the PD-1_HA_ ectodomain served as a much more efficient competitive inhibitor of the hPD-1-biotin/hPD-L1 interaction than did the wild-type hPD-1 ectodomain (Figure 4B).

An analysis of the kinetics of reproduction showed that the total virus yield at 48–120 hpi was on average 1.5–2.0-, 2.4–2.5-, 3.8–4.5-, 4.7–5.0-, and 13.1–19.6-fold lower for OAds with IIIa-EGFP, IIIa-IgGκ-ecto-mPD1, IIIa-IgGκ-ecto-hPD1, IIIa-IgGκ-ecto-hPD1_HA_, and IIIa-Fluc insertions, respectively, than for the parental virus Ad5Δ24RGD (Figure 4C). However, in human cells, the cytolytic efficacy was considerably reduced only for Ad5Δ24RGD_IIIa-EGFP (L3-23K) and only in A549 cells (4.4-fold higher IC_50_ value), whereas no substantial differences were observed between the parental virus and OAds producing the mPD-1 or hPD-1_HA_ ectodomains or EGFP in murine glioma cells (Figure 4D). Finally, we found that the insertion of not only IIIa-IgGκ-ecto-hPD1 but also IIIa-IgGκ-ecto-mPD1 considerably increased the size of plaques in A549 cells but had no effect on the size of plaques in LN18 cells (Figure 4E).

Overall, we provide evidence that the insertion of IIIa-IgGκ-ecto-mPD1, IIIa-IgGκ-ecto-hPD1, or IIIa-IgGκ-ecto-hPD1_HA_ with a high GC3% downstream of the L3-23K region of the E3 region non-deleted Ad5Δ24RGD at a site corresponding to nucleotide 22,354 of the wild-type Ad5 genome did not compromise the cytolytic and plaque formation efficacy.

## 3. Discussion

A transcriptomic analysis of human lung fibroblast IMR-90 cells infected with Ad2 [31] or A549 cells and human fetal lung fibroblast MRC-5 cells infected with Ad5 [32,33,34] revealed that the number of transcripts encoding the fiber protein was substantially greater than that encoding the L3-23K protease at 24, 36 or 48 hpi. However, we found that the protein production/activity of transgenes inserted downstream of the L3-23K region was considerably higher than that of transgenes inserted downstream of the L5-Fiber region, regardless of the transgene length and GC3%. The insertion of a granulocyte-macrophage colony-stimulating factor coding sequence (GM-CSF/CSF2) with a consensus SA sequence downstream of the L3-23K region of Ad5, in which the E1A promoter was replaced by the SV40 poly(A)-human E2F-1 promoter cassette, had no significant effect on the total virus yield or the cytolytic efficacy in A549 cells [16]. In contrast, we found that the insertion of transgenes downstream of the L3-23K region substantially reduced the total yields of Ad5Δ24RGD armed with IIIa-EGFP, IIIa-Fluc, IIIa-IgGκ-ecto-h/mPD1 or SA40-p19FLAG but not with SA40-hPH20. However, a decrease in the total virus yield did not result in a deterioration of the cytolytic efficacy in most cases. Moreover, the plaque formation efficacy was not dependent on the cytolytic efficacy or the total yield of OAds.

*EGFP* and luciferase transgenes inserted downstream of the L5-Fiber region of the wild-type Ad5 and ICOVIR-15K genomes, respectively, decreased the cytolytic efficacy in A549 cells [15,19]. In support of these findings, we observed that the insertion of *EGFP* and *Fluc* not only downstream of the L5-Fiber region but also downstream of the L3-23K region reduced the cytolytic efficacy and kinetics of plaque formation in A549 and LN18 cells. The total yields of OAds with reporter transgenes inserted downstream of the L5-Fiber region were not substantially different from those of the parental virus Ad5Δ24RGD.

GL261 and CT-2A are the most commonly used cells to model syngeneic gliomas in immunocompetent mice [35,36]. We found that the activity of Fluc inserted downstream of the L3-23K region was substantially higher than that of Fluc inserted downstream of the L5-Fiber region in mouse GL261 and CT-2A cells, while it was considerably lower than that in human cells. The Ad5Δ24RGD genome was replicated in GL261 and CT-2A cells, although with considerably lower efficiency than in human tumor cells [21]. Nevertheless, no infectious progeny were produced [21,37], since GL261 and CT-2A cells infected with OAds express low [37] or no detectable [38,39] levels of the fiber protein and, probably, other late proteins. Interestingly, murine diffuse intrinsic pontine glioma XFM and NP53 cells were reported to be semipermissive to Ad5Δ24RGD progeny production and to support fiber protein expression [40]. The infection of murine carcinoma cells with Ad5Δ24-ΔE3B also resulted in a failure to produce infectious virions due to the almost total absence of late structural protein expression despite the high efficacy of hexon gene transcription [41]. Depending on the MOI, the number of transcripts of the hexon gene in murine CMT64 lung carcinoma cells infected with Ad5Δ24-ΔE3B was comparable to or exceeded that in human ovarian carcinoma OVCAR4 cells [41]. Overall, although Ad5 structural proteins may not be produced in murine cells, including GL261 and CT-2A, the transcriptional activity of late genes is retained, which could be exploited to express a therapeutic transgene under the control of the MLP, although this expression is expected to be at a lower level than that in human cells.

Increased hyaluronan (hyaluronic acid) deposition in tumor tissue may prevent the spread of OAds. The delivery of the hyaluronidase hPH20 to tumor tissue has proven useful for improving OAd therapeutic effectiveness in diverse in vivo models [42,43,44,45,46,47,48]. Nevertheless, compared with the parental virus, ICOVIR-15K armed with IIIa-hPH20 or 40SA-hPH20 downstream of the L5-Fiber region displayed slightly decreased cytolytic efficacy (3.8- and 3.2-fold higher IC_50_ values, respectively) in A549 cells [15]. We found that, compared with those of the parental virus, the total virus yield, cytolytic efficacy, and plaque size were similar to or improved (depending on the cell line) for Ad5Δ24RGD_40SA-hPH20 (L5-Fiber). Moreover, although Ad5Δ24RGD_40SA-hPH20 (L3-23K) formed plaques of a smaller size, no considerable differences in the total virus yield or cytolytic efficacy were observed compared with those of the parental virus.

The p19 protein of tomato bushy stunt virus suppresses the RNA interference pathway by sequestering small RNA duplexes and impairing their loading into an RNA-induced silencing complex (RISC) in diverse host systems, including worms, insects, and humans [49,50]. The expression of p19 under the control of the MLP by connecting its cDNA to the fiber coding sequence via a spacer and internal ribosomal entry site (IRES) sequence in the Ad5∆E3 genome improved replication and the production of viral particles in HEK293 cells and increased the killing potency in different cancer cell lines [51]. However, as a control virus, the authors used Ad5∆E3 with *Fluc* instead of *p19* and with the additional modification *E1A*∆24 [51]. We do not know whether the E1A∆24 modification itself had any effect on each specific cell line tested, but we can certainly now assume, in light of our data and the data of Farrera-Sal et al. [15], that a Fluc transgene may not be an adequate control for a p19 transgene, which does not allow us to draw a definitive conclusion about the actual effectiveness of the *p19*-armed Ad5∆E3 in that study [51]. Moreover, a subsequent analysis revealed that p19 is not a universal enhancer of the oncolytic potency of Ads. A comparison of the *E1A*Δ24-modified Group C OAds Ad1-E1AΔ24, Ad2-E1AΔ24, and Ad6-E1AΔ24 with the *p19*-armed OAds Ad1-E1AΔ24-p19, Ad2-E1AΔ24-p19, and Ad6-E1AΔ24-p19, respectively, in A549 and PANC-1 cells demonstrated that p19 production substantially enhanced the cytolytic efficacy of only Ad2-E1AΔ24-p19 and only in PANC-1 cells [52]. We found that the insertion of 40SA-p19FLAG downstream of the L5-Fiber region improved the cytolytic efficacy in A549 and LN18 cells and increased the size of plaques in A549 cells without affecting the total virus yield in either cell line. In contrast, the insertion of 40SA-p19FLAG downstream of the L3-23K region considerably reduced the total virus yield in A549 and LN18 cells and the size of plaques in LN18 cells, while the cytolytic efficacy was similar to that of the parental virus in both lines.

The interaction between PD-1 and PD-L1 results in the inhibition of T-cell effector functions such as cytotoxicity, cytokine release, proliferation, and survival. A soluble mPD-1 ectodomain was proven to inhibit the mPD-1/mPD-L1 interaction and increase antitumor immunity both in vitro and in vivo [53]. We found that the insertion of IIIa-IgGκ-ecto-hPD1 with a GC3% of 83.1% of the coding sequence downstream of the L5-Fiber region did not affect the oncolytic potency of Ad5Δ24RGD in any of the assays. These data contrast previous observations that a high GC3% of a transgene inserted downstream of the L5-Fiber region interferes with oncolytic virus fitness [15,19]. Moreover, the insertion of IIIa-IgGκ-ecto-mPD1, IIIa-IgGκ-ecto-hPD1, or IIIa-IgGκ-ecto-hPD1_HA_ downstream of the L3-23K region, which resulted in substantially higher transgene production than the L5-Fiber region, did not affect the cytolytic efficacy but moderately reduced the total virus yield. Strikingly, the insertion of IIIa-IgGκ-ecto-PD1 downstream of the L3-23K or L5-Fiber region increased the size of plaques in A549 cells. Overall, our data suggest that a high GC3% of a transgene inserted downstream of the L3-23K or L5-Fiber region is not a strict factor determining virus fitness; obviously, additional undefined factors play a role. In support of this suggestion, the cytolytic efficacy of Ad5 armed with *EGFP* with a GC3% of 44.8% or ≈60% (codon deoptimization) inserted downstream of the L5-Fiber region did not differ from that of wild-type Ad5 in A549 cells [19], whereas ICOVIR-15K armed with luciferase with a GC3% of 57.1% showed considerably decreased cytolytic efficacy (140-fold higher IC_50_ value) compared with the parental virus in A549 cells [15]. Thus, GC3% content should not be perceived as an obligatory prerequisite in the design of armed OAds. A thorough analysis of the oncolytic properties of an armed OAd is needed in each case before deciding to optimize/deoptimize codon usage of a transgene. However, it should also be taken into account that codon optimization can have various effects. Accumulated data indicate that synonymous codon substitutions/usage can affect transcription, pre-mRNA splicing, mRNA stability, translation initiation, translation elongation rate, translational fidelity, protein folding, aggregation propensity, sensitivity to limited proteolysis, activity, and secretion [54,55,56].

The optimization of transgene codon usage (GC3%) was proposed as a strategy to preserve virus fitness by reducing transgene–viral intergenic competition [15,19]. Since GC3% content of a transgene is not a sufficient factor for predicting virus fitness, what could be key to understanding why transgenes with similar or relatively high GC3% under the control of the MLP have different effects on virus fitness? A study in *Escherichia coli* showed that competition between overexpressed heterologous coding sequences and the host’s demands for transfer RNA and ribosomal resources affects host fitness [57]. Detailed analysis revealed that the presence of select codons that are overrepresented in native, highly expressed host genes in overexpressed heterologous coding sequences was the cause of the fitness cost [57]. Avoiding specific codons that are overrepresented in such native genes (the Codon Health Index) was proposed as a useful strategy to reduce competition for translation elongation resources and improve fitness [57]. Based on these data, we hypothesize that not GC3% content *per se* but select codons that are overrepresented in highly expressed adenoviral late genes, when present in an overexpressed transgene, might affect virus fitness. This hypothesis requires further experimental testing for transgene-armed oncolytic adenoviruses.

We confirmed that the previously engineered high-affinity hPD-1 ectodomain [30] substantially more effectively inhibited the hPD-1-biotin/hPD-L1 interaction than did the wild-type hPD-1 ectodomain. Given that mPD-L1 interacts with hPD-1 and forms a functional immune checkpoint [58] and that the hPD-1_HA_ ectodomain more potently blocks mPD-L1 on murine B16-F10 cells than does the wild-type hPD-1 ectodomain [30], further in vivo experiments are needed to confirm the efficacy of Ad5Δ24RGD_IIIa-IgGκ-hPD1_HA_ (L3-23K) in murine models, including CT-2A and GL-261 gliomas.

There is evidence that the intratumoral production of the human/mouse PD-1 ectodomain alone or in combination with other therapeutic transgenes may promote the generation of an antitumor immune response. For example, a recombinant myxoma virus encoding the soluble wild-type hPD-1 ectodomain induced antitumor CD8^+^ T-cell responses and tumor eradication in ≈60% of C57BL/6 mice (12/22) bearing subcutaneous B16-F10 melanoma, whereas no animal treated with a control virus displayed a complete response [59]. On the other hand, a replication-incompetent E1/E3-deficient Ad5 armed with the mPD-1 ectodomain fused to the Fc portion of mouse IgG2a and containing an insert of the EF1α promoter in the E3 region did not show any antitumor effect on the subcutaneous murine colon CT26 tumor model in BALB/c mice following an intratumor injection [60]. However, PD-1-Fc in combination with herpes simplex virus thymidine kinase (HSVtk) enzyme expression under the control of the cytomegalovirus (CMV) promoter and a trans-splicing ribozyme targeting human telomerase reverse transcriptase significantly enhanced the regression of not only subcutaneous CT26 tumors in BALB/c mice, but also subcutaneous E.G7-OVA (a derivative of lymphoma EL4 cells stably expressing ovalbumin) tumors in C57/BL6 mice compared with Ad5 encoding the HSVtk enzyme alone [60]. In another study, the E1/E3-deficient Ad5 genome was used as a platform to insert a CMV-EGFP-p2A-E1A expression cassette (a control OAd) and CMV-mPD1 or CMV-mPD1-linker-mCD137L expression cassettes encoding the soluble mPD-1 ectodomain or the ectodomains of mPD-1 and mCD137L fused via a linker, respectively [61]. The survival of the mice treated with a control OAd or the OAd encoding the mPD-1 ectodomain was comparable to that of the mice that received saline in the ascitic murine hepatocellular H22 carcinoma model. In contrast, OAd encoding with the mCD137L ectodomain significantly prolonged survival with an ≈30% cure rate, whereas a 70% cure rate was observed in the mice treated with OAd encoding a fusion of the mPD-1 and mCD137L ectodomains [61]. Thus, an OAd encoding the PD-1 ectodomain in combination with other therapeutic transgenes might be a viable strategy for enhancing therapeutic efficacy and is the subject of further research.

## 4. Materials and Methods

### 4.1. Cell Lines

Human embryonic kidney HEK293, lung adenocarcinoma A549, glioblastoma LN18 (ATCC), and murine glioma CT-2A (#SCC194, Sigma, St. Louis, MO, USA) and GL261 cells (DSMZ Cell Culture Collection, Braunschweig, Germany) were grown in DMEM with L-alanyl-glutamine, 4.5 g/L glucose, and Na pyruvate (#C415, Paneco, Moscow, Russia) supplemented with 10% fetal bovine serum (FBS, HyClone/Cytiva, Marlborough, MA, USA) and penicillin-streptomycin solution at a final concentration of 50 U/mL/50 µg/mL (#A063, Paneco). Cells were maintained in an incubator at 37 °C in a humidified atmosphere of 95% air and 5% CO_2_ and tested negative for mycoplasma contamination (#MR001, MycoReport, Evrogen, Moscow, Russia).

### 4.2. Construction of Oncolytic Adenoviruses

Total RNA was extracted with a phenol:chloroform:isoamyl alcohol mixture (25:24:1) (#77617, Sigma) according to the manufacturer’s instructions. For cloning mouse and human PD-1 into the lentiviral plasmid pLenti-SV40-puro, cDNA was prepared from C57BL/6 mouse spleen or a mixture of human lymphoma cell lines (Jurkat T cells and B-cell non-Hodgkin Burkitt’s lymphomas), respectively, using the High-Capacity cDNA Reverse Transcription kit (#4374966, Applied Biosystems, Waltham, MA, USA). Lentiviral plasmids were used as PCR templates to generate IIIa-IgGκ-ecto-mPD1, IIIa-IgGκ-ecto-hPD1, and IIIa-IgGκ-ecto-hPD1_HA_ products for homologous recombination (Appendix A). The plasmids pCMV6-hPH20/SPAM1 (#SC328840, OriGene, Rockville, MD, USA) and pEGB SF-35S:Renilla:TNOS-35S:P19:TNOS (GB0160) (Addgene, #75412, Watertown, MA, USA) served as templates to generate 40SA-hPH20/SPAM1 and 40SA-p19FLAG products, respectively, for homologous recombination (Appendix A). Phusion Green Hot Start II High-Fidelity PCR Master Mix (#F566L, Thermo Scientific, Waltham, MA, USA) was used to amplify DNA fragments. Before electroporation, gel-extracted PCR products were desalted using membrane filters (#VSWP02500, Millipore, Burlington, MA, USA). A selection/counterselection *rpsL-neo* cassette (#K002, Gene Bridges, Heidelberg, Germany) and λ-Red-mediated linear-circular homologous recombination in the *Escherichia coli* GB08-red strain (#K009, Gene Bridges) were exploited for modifications of the Ad5∆24RGD genome, as detailed previously [21]. Depending on their length, the oligonucleotides were ordered from the DNA synthesis services of Evrogen or Syntol (Moscow, Russia) as high-performance liquid chromatography (HPLC)- or polyacrylamide gel electrophoresis (PAGE)-purified for recombination and site-directed mutagenesis or unpurified for screening colony PCR and sequencing (Appendix A). The viral genomic integrity and the presence of correct target modifications were confirmed by diagnostic digests with EcoRV (#ER0301, Thermo Scientific) and XhoI (#ER0695, Thermo Scientific), and sequencing (Evrogen), respectively. The recombinant adenovirus genomes linearized with PacI (#RK21171, ABclonal, Woburn, MA, USA) were transfected with Lipofectamine 3000 (#L3000015, Invitrogen, Waltham, MA, USA) and rescued in HEK293 cells. The viruses were amplified in A549 cells.

### 4.3. Adenovirus Particle Purification

The crude cell lysate solutions (after three cycles of freeze–thawing and clearing by centrifugation at 2000× *g* for 5 min) or purified viral particles (after two rounds of CsCl density gradient ultracentrifugation) were used for in vitro and in vivo assays (Appendix A). For the prevention of interparticle aggregation of the Ad5∆24RGD-derived viruses during CsCl density gradient ultracentrifugation due to the presence of two free thiol groups of cysteine residues in the RGD-4C peptide inserted in the HI loop of the fiber knob domain [62], the CsCl density solutions (1.27 g/cm^3^ and 1.41 g/cm^3^ ρCsCl in 20 mM HEPES buffer, pH 7.6, 150 mM NaCl) and HEPES buffer for the sample overlay were bubbled with argon gas to cause deoxygenation immediately prior to ultracentrifugation [63]. The collected virus-CsCl bands were dialyzed in virus storage buffer (5 mM Tris, 75 mM NaCl, 1 mM MgCl_2_, 5% sucrose [*w*/*v*], 0.005% polysorbate-80, pH 8.0) using Slide-A-Lyzer 10K dialysis cassettes (#66380, Thermo Scientific). The viruses were aliquoted and stored at −70 °C.

### 4.4. Adenovirus Titration

For determination of an infectious unit titer per milliliter (IFU/mL), OAds were titered on A549 cells in DMEM/10% FBS. The cells were seeded in 48-well plates (1.15 × 10^5^ cells/well, 250 µL) and infected in suspension with 50 or 100 µL of the serially diluted viral stock solutions. Two days later, the cell monolayers were fixed with 100% ice-cold methanol for 10 min at −20 °C, stained with a polyclonal anti-adenovirus type 5 antibody (1:3000, ab6982, Abcam, Cambridge, UK) and a goat anti-rabbit immunoglobulin G (IgG) H&L (horseradish peroxidase [HRP]) (1:1000, ab6721, Abcam) for 45 min each at 37 °C with agitation, and then visualized with a 3,3′-diaminobenzidine (DAB) substrate kit (ab64238, Abcam). Stained cells were counted under a light microscope (100× magnification) in 10 random fields/well. The viral infectious titer was calculated using the following formula: IFU/mL = [(average positive cells/field) × (fields/well)]/[volume virus (mL) × dilution factor]. The mean IFU/mL values of at least two independent titrations were used for the assays (Appendix A).

### 4.5. Resazurin/Alamar Blue Cell Viability Assay

Human tumor cells (5 × 10^3^/well) and rodent glioma cells (2.5 × 10^3^/well) were seeded in 80 µL of DMEM/10% FBS in 96-well plates and infected in suspension (20 µL of virus inoculum) with serial 3-fold dilutions of the virus stocks in DMEM/5% FBS. The next day, 100 µL of additional DMEM/10% FBS was added to each well. Five days post-infection, the medium was replaced with 100 µL of DMEM/10% FBS/10% resazurin. A 0.15 mg/mL solution of resazurin sodium salt (#AC41890-0010, Acros Organics, Geel, Belgium) in Dulbecco’s phosphate-buffered saline (DPBS, P060, Paneco) was used (filter-sterilized, aliquoted for single use, and stored at −20 °C). After 4 h of incubation, the fluorescence emission was measured with an EnSpire multimode microplate reader (PerkinElmer, Waltham, MA, USA).

### 4.6. Total Virus Yield Assay

The cells (1 × 10^5^/well) were seeded in 250 µL of DMEM/10% FBS in 48-well plates and infected in suspension (100 µL of virus inoculum, DMEM/5% FBS) at an MOI of 10 IFU/cell for A549 cells and 50 IFU/cell for LN18 cells. Both the supernatants along with the cells were collected at the indicated time points. After three cycles of freeze–thawing, the IFU/mL titers were determined by immunocytochemistry in A549 cells as described above.

### 4.7. Plaque Assay

For preparation of 1L of sterile filtered 2× DMEM, a powder with glutamine (#12800017, Gibco, Waltham, MA, USA) was dissolved in deionized autoclaved water, supplemented with 100 mL of 7.5% sodium bicarbonate (#25080060, Gibco), and penicillin-streptomycin solution at a final concentration of 100 U/mL/100 µg/mL. The cells were seeded in 6-well plates (6 × 10^5^ cells/well for A549 and 9 × 10^5^ for LN18) and infected at ≈80–90% confluence with serial dilutions of viruses the next day. The appropriate dilution range for each cell line should be determined empirically to ensure that plaques are sufficiently spaced apart and do not merge (e.g., 0.01–0.0001 IFU/cell as a starting point). Two to three hours post-infection, the virus inoculum was removed, and the cell monolayers were covered with 2 mL of a mixture of 1× DMEM/5% FBS/1% agarose I (Biotechnology grade, Am-0710, VWR International, Radnor, PA, USA). After 1× DMEM/5% FBS/1% agarose solidified in a plate, the DMEM/5% FBS overlay was added to each well. Fresh DMEM/5% FBS was added as needed throughout the assays. For evaluation of MTT-stained plaques, the aqueous medium overlay was aspirated and the cell monolayers covered with agarose were stained by incubation with a 1/10th volume of MTT (0.5 mg/mL at a final concentration) for 3–4 h at 37 °C in a CO_2_ incubator. The fluorescent and MTT-stained plaques were photographed on the indicated days at 50× magnification with a Leica DM3000 microscope (Wetzlar, Germany) and quantified with Fiji ImageJ (https://imagej.net/software/fiji/, last accessed on 1 March 2025). Due to significant variations in the size and morphology of plaques, the plaque contours were manually traced using the tool “freehand selections”, while the plaque areas (µm^2^) were calculated automatically after setting the size of the scale bar of images as a reference for the actual size (set scale menu in the analyze tab).

### 4.8. Flow Cytometry

The cells (5 × 10^5^/well) were infected in suspension with serial dilutions of viruses in 12-well plates in a total volume of 1 mL of DMEM/10% FBS. On the indicated days of analysis, the cells were detached by treatment with trypsin (#P039, Paneco) and washed with cold flow cytometry buffer consisting of PBS (#4417-100TAB, Sigma) supplemented with 2 mM ethylenediaminetetraacetic acid disodium salt dihydrate (Na_2_EDTA × 2H_2_O, #11280, Serva, Heidelberg, Germany) and 1% bovine serum albumin (BSA, #0332-100G, Amresco, Solon, OH, USA). Data were acquired via a MoFlow XDP (Beckman Coulter, Brea, CA, USA) and analyzed using Summit V5.2 (Beckman Coulter). At least 1 × 10^4^ events were analyzed.

### 4.9. In Vitro Bioluminescence Assay

The cells (2.5 × 10^4^/well) were seeded in 80 µL of DMEM/10% FBS in 96-well plates and infected in suspension (20 µL of virus inoculum) with serial 3-fold dilutions of virus stocks in DMEM/5% FBS. Twenty-four and forty-eight hpi, the medium was aspirated, and D-luciferin potassium salt (30 mg/mL in PBS, #122799, PerkinElmer) diluted 1:200 in DMEM/5% FBS (150 µg/mL final concentration) was added (100 µL/well) to the cells 10 min before imaging. Luminescence (relative light unit [RLU]) was measured with an EnSpire multimode microplate reader (Austin, TX, USA).

### 4.10. Enzyme-Linked Immunosorbent Assay (ELISA)

To analyze the concentrations of hPD-1 and hPD-1_HA_ ectodomains in the supernatants, the cells (1 × 10^5^/well) were seeded in 48-well plates and infected in suspension at an MOI of 5 IFU/cell (A549), 50 IFU/cell (LN18 and CT-2A), or 100 IFU/cell (GL261) in DMEM/5% FBS. The supernatants were collected at 48 and 72 hpi, cleared for 5 min at 2000× *g*, at 4 °C, and stored at −70 °C. The supernatants were analyzed using the human PD-1 ELISA kit (ab252360, Abcam) according to the manufacturer’s instructions.

To analyze the competitive antagonist activity of hPD-1 and hPD-1_HA_ ectodomains, A549 cells (5 × 10^6^) were infected in suspension at an MOI of 20 IFU/cell in a total volume of 0.5 mL of DMEM/1% FBS at 37 °C in a CO_2_ incubator for 1 h, pipetting the cells periodically to ensure uniform infection. Then the cells with virus inoculums were seeded onto T75 flasks in a total volume of 10 mL of DMEM/1% FBS. Seventy-two hpi, the conditioned medium was collected and cleared for 5 min at 2000× *g*, 4 °C. The supernatants were then concentrated 20-fold in 15 mL centrifugal filters, 5kDa (#UTPES150005N, Membrane Solutions, Shanghai, China), aliquoted, and stored at −70 °C. The concentrated supernatants were analyzed using the PD-1 [Biotinylated]:PD-L1 inhibitor screening ELISA kit (#EP-101-96tests, ACROBiosystems, Newark, DE, USA). Recombinant human PD-1 protein (active) (ab174035, Abcam) was used as a control.

### 4.11. Hyaluronidase Activity Assay

A549 cells (1 × 10^5^/well) were infected in suspension at an MOI of 10 IFU/cell in a total volume of 350 µL of DMEM/10% FBS in 48-well plates. The supernatants were collected at 48 hpi, cleared by centrifugation at 2000× *g* for 5 min, 4 °C, and stored at −70 °C. Alternatively, A549 cells (3 × 10^6^) were infected in suspension at an MOI of 20 IFU/cell in a total volume of 0.5 mL of DMEM/1% FBS at 37 °C in a CO_2_ incubator for 1 h, pipetting the cells periodically to ensure uniform infection. Then, the cells inoculated with the virus were seeded into T25 flasks in a total volume of 5 mL of DMEM/1% FBS. Forty-eight hpi, the conditioned medium was collected and cleared at 2000× *g* for 5 min at 4 °C. The supernatants were then concentrated 15-fold in 15 mL centrifugal filters, 5 kDa (#UTPES150005N, Membrane Solutions), aliquoted, and stored at −70 °C. A KRISHZYME hyaluronidase enzymatic assay kit (#KBBA05, Krishgen BioSystems, Mumbai, India; calibration range 0–28 U/mL for lot #HYA0922) was used to measure hPH20 activity in accordance with the manufacturer’s protocol. A turbidimetric reaction at 600 nm was measured in a CLARIOstar Plus microplate reader (BMG Labtech, Ortenberg, Germany).

### 4.12. Western Blot Analysis

A549 cells (1 × 10^6^) were infected at an MOI of 5 IFU and cultured in DMEM/5% FBS for 36 h prior to cell lysis. Cells were washed once with PBS and lysed with RIPA Lysis and Extraction Buffer (#89900, Thermo Scientific) containing cOmplete Protease Inhibitor Cocktail (#11697498001, Roche, Basel, Switzerland). Twelve micrograms of the total protein were separated in 12% SDS-PAGE gels and transferred onto PVDF membranes (0.45 µm, Amersham Hybond P). Membranes were blocked for 1 h at room temperature in 4% non-fat milk/TBST (0.05% Tween-20, TBS, Atlanta, GA, USA) and incubated with primary antibodies in 2% non-fat milk overnight at 4 °C. The primary antibodies were mouse monoclonal horseradish peroxidase (HRP)-conjugated anti-Flag (1:2000, ab49763, Abcam) and anti-β-actin (1:3500, sc-47778, SantaCruz, Dallas, TX, USA). Mouse IgG kappa binding protein (m-IgGκ BP) conjugated to HRP (1:3000, sc-516102, SantaCruz) was used for binding to anti-β-actin immunoglobulins. The membranes were incubated with Clarity Western ECL Substrate (#1705060, Bio-Rad Laboratories, Hercules, CA, USA) according to the manufacturer’s instructions. The chemiluminescent signal was detected using ChemiDoc MP (Bio-Rad).

### 4.13. In Vivo Bioluminescence Imaging (IVIS)

The experimental procedures were performed in accordance with Directive 2010/63/EU of 22 September 2010 on the protection of animals used for scientific purposes and were approved by the local ethical committee of V.P. Serbsky National Medical Research Center. Eight-week-old female C57BL/6 mice obtained from the Scientific Center of Biomedical Technologies of the Russian Academy of Science (Andreevka) were maintained in individually ventilated cages (Tecniplast, Milan, Italy). The mice (*n* = 5 per group) were anesthetized via the intraperitoneal injection of 20 mg/kg Zoletil (Virbac, Carros, France) with 5 mg/kg xylazine (Nita-Farm, Saratov, Russia). Murine CT-2A glioma cells were pre-infected with Ad5Δ24RGD_IIIa-Fluc (L3-23K) at an MOI of 50 IFU/cell in suspension for 2 h at 37 °C in a CO_2_ incubator, pipetting the cells periodically, then washed, and 5 µL of cell suspension (total 5 × 10^5^ cells) in DMEM was delivered into the right hemisphere at coordinates 1.5 mm anterior, 2.5 mm lateral from bregma, and 3.5 mm ventral below the skull surface with a 26-gauge needle at a rate of 1 µL/min using a digital stereotaxic instrument (RWD) and an R462 syringe pump (RWD). In another experiment, a single intratumoral stereotactic injection of the virus (5 × 10^8^ IFU in 5 µL of storage buffer) was carried out 14 days after intra-brain inoculation of 5 × 10^4^ CT-2A cells. Quantitative analysis of the bioluminescence signals (radiance) was performed using an IVIS Spectrum system (PerkinElmer) at days 1, 2, 3, and 6. The animals received D-luciferin potassium salt (#122799, PerkinElmer) intraperitoneally at a dose of 3 mg/animal (≈20 g weight) 10–15 min before imaging. The heads of the mice were not shaved before imaging. IVIS parameters for imaging were exposure time: 30 s; binning: 8 (medium); f/stop (lens aperture): 1.

### 4.14. Statistics

The plotted IC_50_ values (IFU/cell) were estimated from dose–response nonlinear regression curves ([inhibitor] versus normalized response—variable slope) using GraphPad Prism v8 software.

## Figures and Tables

**Figure 1 ijms-26-03700-f001:**
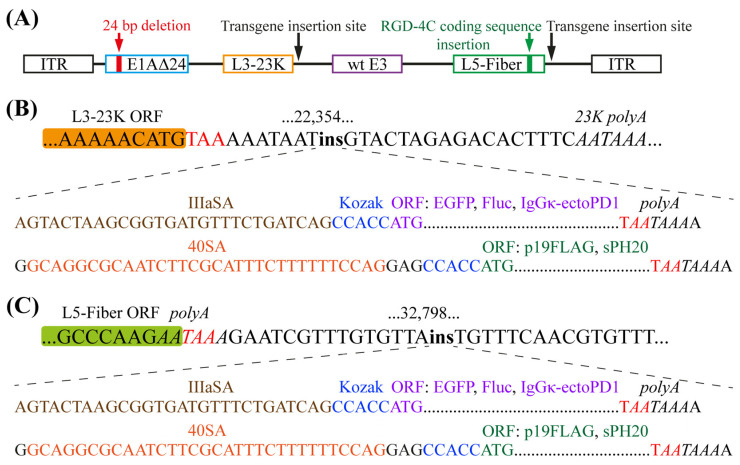
Construction of transgene-armed OAds. (**A**) Schematic representation of Ad5Δ24RGD genome modifications. Delta-24, or Δ24, is a 24-base pair deletion in a sequence encoding the conserved region 2 (CR2) domain of the E1A protein. RGD-4C is an arginine–glycine–aspartic acid (RGD) motif-containing αVβ3 and αVβ5 integrin-targeting peptide inserted in the HI loop of the fiber knob domain to improve the infection of coxsackievirus and adenovirus receptor (CAR)-negative/low cells. Ad5∆24RGD was used as a platform to incorporate transgenes downstream of the L3-23K and L5-Fiber regions. ITR, inverted terminal repeat. (**B**) A transgene open reading frame (ORF) was inserted into a cassette containing a splicing acceptor (SA) of the Ad40 long fiber gene (40SA for *p19*FLAG and *hPH20*) or an SA of the Ad5 pIIIa gene (IIIaSA for *EGFP*, *Fluc*, and IgGκ-ecto-hPD1), a Kozak consensus sequence, and a polyadenylation (polyA) signal sequence overlapping with a stop codon downstream of the L3-23K region at a site corresponding to nucleotide 22,354 of the wild-type Ad5 genome. (**C**) A transgene ORF was inserted into a cassette containing IIIaSA or SA40, a Kozak consensus sequence, and a polyA signal sequence downstream of the L5-Fiber region at a site corresponding to nucleotide 32,798 of the wild-type Ad5 genome.

**Figure 2 ijms-26-03700-f002:**
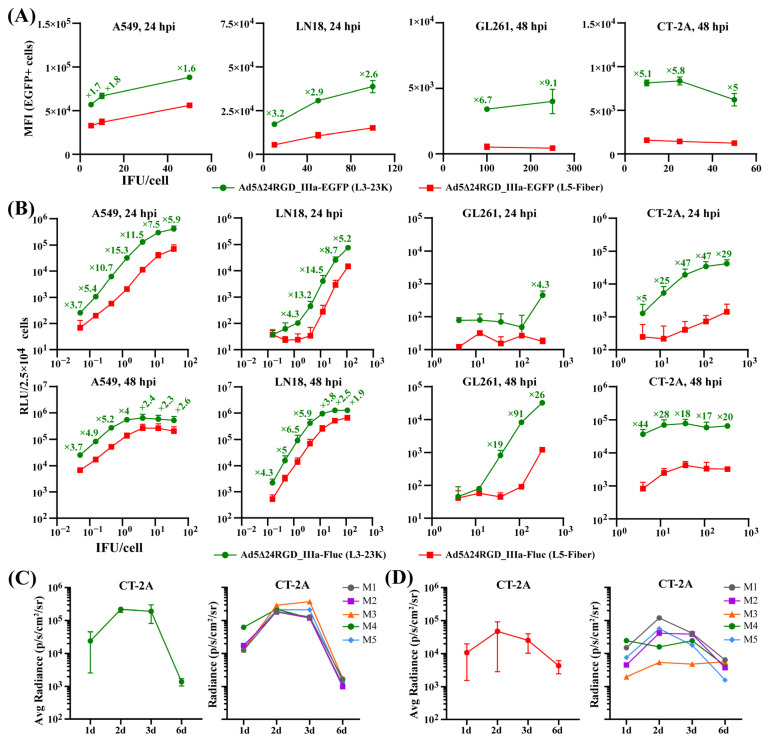
The production of a transgene inserted downstream of the L3-23K region is higher than that of a transgene downstream of the L5-Fiber region in human and mouse tumor cells. (**A**) Cells (5 × 10^5^/well) were infected in suspension with the indicated *EGFP*-armed OAds at a multiplicity of infection (MOI) of 5, 10, and 50 IFU/cell for A549 cells, 10, 50, and 100 IFU/cell for LN18 cells, 100 and 250 IFU/cell for GL261 cells, and 10, 25, and 50 IFU/cell for CT-2A cells, and the median fluorescence intensity (MFI) was analyzed at 24 or 48 hpi by flow cytometry. The data are shown as means (SD) of two independently repeated experiments (*N* = 2). (**B**) Cells (2.5 × 10^4^/well) were infected in suspension with a serial 3-fold dilution of the indicated *Fluc*-armed OAds starting from an MOI of 33 IFU/cell for A549 cells, 100 IFU/cell for LN18 cells, and 333 IFU/cell for GL261 and CT-2A cells. Luminescence (relative light units [RLUs]) was analyzed at 24 and 48 hpi. The means (SD) of two independently repeated experiments (*N* = 2) with four technical replicates are shown. Only ≥2-fold differences in the mean RLU values are indicated. (**C**) CT-2A glioma cells were preinfected at an MOI of 50 IFU/cell in suspension in serum-free DMEM for 2 h, washed, and injected into the brains (5 × 10^5^ cells/5 µL) of syngeneic C57BL/6 mice (*n* = 5). Luminescence (radiance [p/s/cm^2^/sr]) was measured on the indicated days posttreatment. The mean luminescence (average radiance) per treatment group (SD) and the individual measurements of luminescence (radiance) for each animal are plotted. (**D**) C57BL/6 mice (*n* = 5) bearing CT-2A tumors were injected intratumorally with the virus at a dose of 5 × 10^8^ IFU/5 µL at 14 days after the intrabrain inoculation of tumor cells (5 × 10^4^ cells/3 µL), and luminescence (radiance) was measured on the indicated days posttreatment. The mean luminescence (average radiance) per treatment group (SD) and the individual measurements of luminescence (radiance) for each animal are plotted. See also Appendix A.

**Figure 3 ijms-26-03700-f003:**
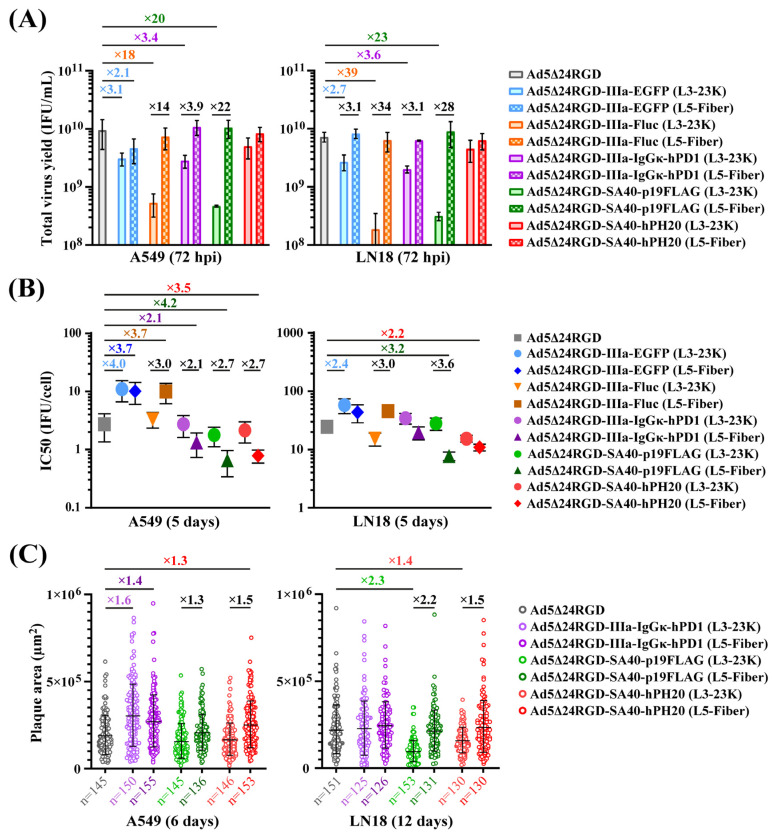
The insertion of transgenes downstream of the L3-23K or L5-Fiber region differentially and unpredictably affects the oncolytic potency of Ad5Δ24RGD. (**A**) A comparative analysis of the total virus yields was performed by harvesting the cell lysates along with the supernatants at 72 hpi. A549 and LN18 cells (1 × 10^5^/well) were infected in suspension with the indicated OAds at an MOI of 10 IFU/cell or 50 IFU/cell, respectively. The titer (infectious units per milliliter, IFU/mL) was determined using immunocytochemistry with anti-capsid staining. The data are shown as means (SD) of two independently repeated experiments (*N* = 2). Only ≥2-fold differences in the mean total virus yields are indicated. (**B**) Comparison of the IC_50_ values obtained from viability curves after the infection of cells (5 × 10^3^/well) in suspension in triplicate with a serial 3-fold dilution of the indicated OAds starting from an MOI of 333 IFU/cell for A549 cells or 1000 IFU/cell for LN18 cells. The data were collected on Day 5 postinfection. The data are shown as means (SD) of three independently repeated experiments (*N* = 3). Only ≥2-fold differences in the mean IC_50_ values are indicated. (**C**) Comparison of the sizes of MTT-stained plaques in the cancer cell monolayers under the agarose overlay. The cells were seeded in 6-well plates (6 × 10^5^ cells/well for A549 and 9 × 10^5^ for LN18) and infected the following day with appropriate dilutions of viruses determined empirically (e.g., 0.01–0.0001 IFU/cell as a starting point). The sizes of 40–60 random plaques from two to three wells were taken for analysis from each independent experiment. The total number (*n*) of analyzed plaques collected from three independently repeated experiments (*N* = 3) are indicated for each group. The data are shown as means (SD) of the total number (*n*) of analyzed plaques. Only ≥1.3-fold differences in the mean plaque sizes are indicated. See also Appendix A.

**Figure 4 ijms-26-03700-f004:**
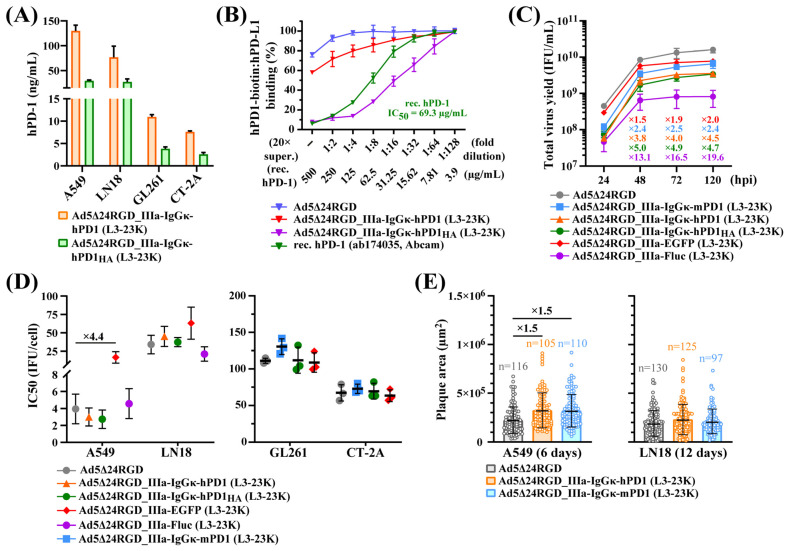
Characterization of Ad5Δ24RGD expressing the wild-type murine PD-1, wild-type human PD-1 or human mutant high-affinity PD-1_HA_ ectodomain. (**A**) Evaluation of the hPD-1 and hPD-1_HA_ ectodomain concentrations in the supernatants of human and mouse cells by an enzyme-linked immunosorbent assay (ELISA). The cells (1 × 10^5^/well) were infected in suspension at an MOI of 5 IFU/cell for A549, 50 IFU/cell for LN18 and CT-2A, and 100 IFU/cell for GL261. The supernatants were collected at 48 hpi. The data are shown as means (SD) of two independently repeated experiments (*N* = 2). (**B**) Comparative analysis of the competitive antagonist activity of the hPD-1 and hPD-1_HA_ ectodomains. At seventy-two hours after the infection of 5 × 10^6^ A549 cells with OAds at an MOI of 20 IFU/cell, the conditioned medium was collected, concentrated 20-fold, and analyzed using a PD-1-biotin/PD-L1 ELISA. The recombinant human PD-1 protein (active) (ab174035, Abcam) was used as a control. The data are shown as means (SD) of two independently repeated experiments (*N* = 2). (**C**) A comparative analysis of the total virus yields was performed by harvesting the cell lysates along with the supernatants at 24, 48, 72, and 120 hpi. A549 cells (1 × 10^5^) were infected in suspension with the indicated OAds at an MOI of 10 IFU/cell. The titer is reported in infectious units per milliliter (IFU/mL) and was determined using immunocytochemistry with anti-capsid staining. The data are shown as means (SD) of three independently repeated experiments (*N* = 3). (**D**) Comparison of the IC_50_ values obtained from viability curves after infection of human cells (5 × 10^3^/well) and mouse cells (2.5 × 10^3^/well) in suspension with serial 3-fold dilutions of the indicated OAds starting from an MOI of 333 IFU/cell for A549 cells, 1000 IFU/cell for LN18 cells, 2000 IFU/cell for GL261, and 666 IFU/cell for CT-2A cells. The data were collected on Day 5 postinfection. Two (human cells) or three (mouse cells) independently repeated experiments with technical triplicates were performed. The data are shown as means (SD) of independently repeated experiments (*N* = 2 − 3). Only ≥2-fold differences in the mean IC_50_ values are indicated. (**E**) Comparison of the sizes of MTT-stained plaques in the cancer cell monolayers under the agarose overlay. The cells were seeded in 6-well plates (6 × 10^5^ cells/well for A549 and 9 × 10^5^ for LN18) and infected the following day with appropriate dilutions of viruses determined empirically. The sizes of 40–70 random plaques from two to three wells were taken for analysis from each independent experiment. The total number (*n*) of analyzed plaques collected from two independently repeated experiments (*N* = 2) are indicated for each group. The data are shown as means (SD) of the total number (*n*) of analyzed plaques. Only ≥1.3-fold differences in the mean plaque sizes are indicated.

## Data Availability

The original contributions presented in this study are included in the article/Appendix A. Further enquiries can be directed to the corresponding author.

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
