# Peer review of "Comparison of the L3-23K and L5-Fiber Regions for Arming the Oncolytic Adenovirus Ad5-Delta-24-RGD with Reporter and Therapeutic Transgenes"

_ijms, 2025, doi:10.3390/ijms26083700_

Round 1

Reviewer 1 Report

Comments and Suggestions for Authors

L3-23K and L5-Fiber are two regions of the adenovirus genome which are under regulation by the major late promoter, thus the proteins expressed from these regions are activated at later stages of infection (after DNA replication has occurred). This study aims to compare these two sites for the insertion of reporter and therapeutic transgenes into the oncolytic adenovirus Ad5-delta-24-RGD. The authors used 5 different transgenes (EGFP, luciferase, p19, PD-1 ectodomains, and hyaluronidase PH20) to assess how the insertion sites affect transgene expression, viral fitness, virus yield, cytolytic efficacy and plaque formation. It was previously hypothesized that a higher percentage of codons with G or C nucleotides at the third base pair position (GC3%) of transgenes inserted in L5-Fiber negatively impacts oncolytic potency. The authors found that inserting transgenes downstream of the L3-23K regions generally led to higher expressions levels but was often accompanied by a decrease of virus yield. Viruses with transgenes inserted in the L5-Fiber region showed lower expression levels and higher virus yields. Notably, the relationship between GC3% and the oncolytic potency of viruses was inconsistent. For example, PD-1 ectodomains with high GC3%, had minimal impacts on cytolytic efficacy and even increased plaque size in a cell-line-specific manner. The findings challenge the notion that GC3% alone dictates viral potency and highlight that no universal rule governs the optimal insertion site. Instead, each transgene should be tested individually to balance therapeutic expression and oncolytic efficacy.

Despite the valuable insight this paper provides on how insertion sites of transgenes influence expression and viral fitness in oncolytic adenoviruses, the underlying mechanisms remain partially explored and some figures raise concerns of potential pseudoreplication.

Major points:

  1. Concerns of pseudoreplication: Many figures mention technical replicates but do not specify if biological replicates were used (independent virus preparations, different cells passages, etc.). In figure 3C, plaques sizes from the same experiment were measured and plotted on a graph, with the number of plaques measured denoted by “n”. This suggest that each individual plaque was treated as an independent replicate which could be considered pseudoreplication if the measurements originated from the same biological sample. This in turn inflates the sample size and makes the data look more robust that it actually is.
  2. Lack of negative controls: The authors compare the expression of various transgenes inserted in the L3 or L5 region of oncolytic Ads. However, some experiments do not consistently use a negative control (such as an empty virus backbone with no transgene). Some examples are figure 2 and figure 4A.
  3. A main point of this manuscript is that GC3% content is not sufficient for predicting viral fitness. I think it would be beneficial to propose alternate hypothesis that can potentially explain discrepancies between expression profiles of different transgenes. Example: transgene toxicity or secondary structures in the RNA.
  4. As mentioned in the article, GC3% of transgenes do not consistently correlate with viral fitness. There is a lack of statistical analyses as to how the GC3% correlates with transgene expression, cytolytic efficacy and plaques sizes. The manuscript would benefit from a graph that compares GC3% to viral fitness indicators such as total virus yield, IC50 values, and plaque sizes.

Minor points:

  1. Lack of rationale for cell line choice: The cell lines used in this paper (A549, LN18, GL261, CT-2A) vary in permissiveness to adenovirus replication. However, there was a lack of explanation as to why these cell lines were used and how they might affect virus replication.
  2. Variability in plaque assay: Plaque size seem to be quite variable with large standard deviations (Figure 3C). It would be beneficial to clarify how plaques were measured (manual counting, ImageJ, Cell Profiler, etc.).
  3. Reporting of experimental conditions are inconsistent and sometimes incomplete. It would be beneficial to ensure that timepoints, MOIs, seeding densities are consistently stated across all figures.
  4. Depending on the GC3% content of the transgenes, the authors switched between the use of 2 different splice acceptors (IIIaSA and 40SA). However, the rational as to why one splice acceptor would work better than the other for a specific transgene is missing.
  5. The discussion section of the article seems to repeat the results section. It would be beneficial to condense the repetitive parts in the discussion.
  6. There seems to be a lack of consistency in regards to the graphical elements. While some figures include bars and numerical indicators of mean fold change, others do not. Some figures indicate the fold change is non-significant (NS) and other figures simply have no indication for this.
  7. The method used for the detection of transgene protein is inconsistent. For some transgenes, ELISA assays were used. For others, flow cytometry is used. For p19 protein expression quantification, only a western blot was conducted. Additionally, this western blot is only present in the supplementary data when it should be included the in the manuscript.

Author Response

Comments #1: L3-23K and L5-Fiber are two regions of the adenovirus genome which are under regulation by the major late promoter, thus the proteins expressed from these regions are activated at later stages of infection (after DNA replication has occurred). This study aims to compare these two sites for the insertion of reporter and therapeutic transgenes into the oncolytic adenovirus Ad5-delta-24-RGD. The authors used 5 different transgenes (EGFP, luciferase, p19, PD-1 ectodomains, and hyaluronidase PH20) to assess how the insertion sites affect transgene expression, viral fitness, virus yield, cytolytic efficacy and plaque formation. It was previously hypothesized that a higher percentage of codons with G or C nucleotides at the third base pair position (GC3%) of transgenes inserted in L5-Fiber negatively impacts oncolytic potency. The authors found that inserting transgenes downstream of the L3-23K regions generally led to higher expressions levels but was often accompanied by a decrease of virus yield. Viruses with transgenes inserted in the L5-Fiber region showed lower expression levels and higher virus yields. Notably, the relationship between GC3% and the oncolytic potency of viruses was inconsistent. For example, PD-1 ectodomains with high GC3%, had minimal impacts on cytolytic efficacy and even increased plaque size in a cell-line-specific manner. The findings challenge the notion that GC3% alone dictates viral potency and highlight that no universal rule governs the optimal insertion site. Instead, each transgene should be tested individually to balance therapeutic expression and oncolytic efficacy.

Authors’ Response #1: We thank the reviewer for a detailed analysis of the manuscript and for the comments formulated in order to improve the quality of the presentation of the results.

Comments #2: Despite the valuable insight this paper provides on how insertion sites of transgenes influence expression and viral fitness in oncolytic adenoviruses, the underlying mechanisms remain partially explored and some figures raise concerns of potential pseudoreplication.

Major points:

Concerns of pseudoreplication: Many figures mention technical replicates but do not specify if biological replicates were used (independent virus preparations, different cells passages, etc.). In figure 3C, plaques sizes from the same experiment were measured and plotted on a graph, with the number of plaques measured denoted by “n”. This suggest that each individual plaque was treated as an independent replicate which could be considered pseudoreplication if the measurements originated from the same biological sample. This in turn inflates the sample size and makes the data look more robust that it actually is.

Authors’ Response #2: Thank you for pointing this out. According to Chan and Teo (Stem Cells. 2020;38(9):1055-1059. doi: 10.1002/stem.3237), “Biological replicates … are independently repeated experiments performed on cells of the same cell line but derived from a biologically distinct source or of a different passage”. In the caption to Figure 3C, we originally stated the following: “Three independent experiments were performed”. For each part of the figures, the number of independently repeated experiments (N) is indicated in the captions.

In plaque assays, plaque sizes for any oncolytic adenovirus in a monolayer of cells of any cell line significantly vary in the same experiment and between independently repeated experiments performed on different days. Therefore, we observe significant variations in the values of plaque sizes in the graphs. The explanations for this phenomenon are given in these papers (DOI: 10.1128/msphere.00454-24; DOI: 10.3390/v13081568). In short, there is significant cell-to-cell variability in adenovirus entry, transcription, and plaque formation. Cell states confer high or low susceptibility of individual human cells to adenovirus infection. The underlying cell state determining infection variability is maintained for at least 9 weeks of cell cultivation after cloning the culture… (DOI: 10.1128/msphere.00454-24; DOI: 10.3390/v13081568) It follows that to accurately compare plaque sizes of two or more oncolytic adenoviruses, it is necessary to collect a sufficient number of random plaques to account for natural variability in cell states (i.e., plaque sizes). To account for variations due to experimental procedures, we performed three independent experiments (N = 3) in which cells were used at different passages. According to Table S4, each individual virus for in vitro assays was represented by only one independent preparation titrated independently at least twice.

As for pseudo-replication. According to Lazic et al. (What exactly is ‘N’ in cell culture and animal experiments? PLoS Biol 16(4): e2005282; doi.org/10.1371/journal.pbio.2005282), “The sample size (N) is equal to the number of experimental units.” “Experimental unit is the entity that is randomly and independently assigned to experimental conditions. “In addition to random and independent assignment, for genuine replication: 1) The treatment(s) should be applied independently to each experimental unit, and 2) The experimental units should not influence each other.” Thus, formally, according to the formulated criteria, we are indeed dealing with pseudo-replication. Taking into account the significant biological variation in plaque sizes, determined by the state of each individual infected cell, we think it is also non-informative to indicate on the graph only three values for each group, obtained as a result of three independently repeated experiments (N = 3), and to calculate statistics based on these three values. Therefore, we left on the graphs the designation of the ratios (fold changes) of the means between groups and removed all statistical data.

Finally, the following description of the plaque assays in the figure captions has been included in the revised version of the manuscript: “The sizes of 40-60 [or another range specific for each cade] random plaques from two to three wells of a 6-well plate were taken for analysis from each independent experiment. The total number (n) of analyzed plaques collected from three [or more] independently repeated experiments (N = 3 [or more]) are indicated for each group. The data are shown as means (SD) of the total number (n) of analyzed plaques.”

Comments #3: Lack of negative controls: The authors compare the expression of various transgenes inserted in the L3 or L5 region of oncolytic Ads. However, some experiments do not consistently use a negative control (such as an empty virus backbone with no transgene). Some examples are figure 2 and figure 4A.

Authors’ Response #3: We did not plot data from cells infected with the parental virus Ad5Δ24RGD because such cells did not show fluorescence above non-infected cells used to collect flow cytometry light scatter, did not emit light in the presence of luciferin (RLU was similar to uninfected cells or empty wells and too low to plot on a logarithmic scale), and did not endogenously produce PD-1 or PH20 hyaluronidase. In the latter two cases, we added a note about the lack of endogenous production in the text of the manuscript.

Comments #4: A main point of this manuscript is that GC3% content is not sufficient for predicting viral fitness. I think it would be beneficial to propose alternate hypothesis that can potentially explain discrepancies between expression profiles of different transgenes. Example: transgene toxicity or secondary structures in the RNA.

Authors’ Response #4: The authors who proposed the GC3% content hypothesis had previously ruled out possible direct toxicity of transgenes as a potential speculative cause. We have added the following discussion to the manuscript: “The optimization of transgene codon usage (GC3%) was proposed as a strategy to preserve virus fitness by reducing transgene‒viral intergenic competition [15,19]. Since GC3% content of a transgene is not sufficient factor for predicting virus fitness, what could be key to understanding why transgenes with similar or relatively high GC3% under the control of the MLP have different effects on virus fitness? A recent study in Escherichia coli showed that competition between overexpressed heterologous coding sequences and the host's demands for tRNA and ribosomal resources affects host fitness (DOI: 10.1126/sciadv.adk3485). Detailed analysis revealed that the presence of select codons that are overrepresented in native, highly expressed host genes in overexpressed heterologous coding sequences was the cause of the fitness cost (DOI: 10.1126/sciadv.adk3485). Avoiding specific codons that are overrepresented in such native genes (the Codon Health Index) was proposed as a useful strategy to reduce competition for translation elongation resources and improve fitness (DOI: 10.1126/sciadv.adk3485). Based on these data, we hypothesize that not GC3% content per se but select codons that are overrepresented in highly expressed adenoviral late genes, when present in an overexpressed transgene, might affect virus fitness. This hypothesis requires further experimental testing for transgene-armed oncolytic adenoviruses.”

Comments #5: As mentioned in the article, GC3% of transgenes do not consistently correlate with viral fitness. There is a lack of statistical analyses as to how the GC3% correlates with transgene expression, cytolytic efficacy and plaques sizes. The manuscript would benefit from a graph that compares GC3% to viral fitness indicators such as total virus yield, IC50 values, and plaque sizes.

Authors’ Response #5: We presented graphically the relationship between GC3% content, amino acid length, and viral fitness indicators (total virus yield, cytolytic activity [IC50], and plaque sizes) of the transgenes inserted downstream of the L3-23K and L5-Fiber regions. This illustration is included in the supplementary data, Figure S3.

Minor points:

Comments #6: Lack of rationale for cell line choice: The cell lines used in this paper (A549, LN18, GL261, CT-2A) vary in permissiveness to adenovirus replication. However, there was a lack of explanation as to why these cell lines were used and how they might affect virus replication.

Authors’ Response #6: A549 cells are highly permissive producers of oncolytic adenoviruses. This line is used for amplification of oncolytic adenoviruses after rescue in HEK293 cells and as a reference in the field of oncolytic adenoviruses. The human glioblastoma LN18 cells and murine glioma GL261 and CT-2A cells were taken for analysis, since the parental adenovirus Ad5∆24RGD has so far been used in clinical trials of phases I/II only for the treatment of brain tumors – gliomas. All these cell lines are routinely used in the experiments of groups developing oncolytic adenoviruses for the treatment of brain tumors. We also note that the original version of our manuscript stated that “Ad5∆24RGD (DNX-2401, tasadenoturev) has been tested in phase I/II clinical trials in patients with high-grade glioma and presented a favorable toxicity profile [24–27].” Therefore, in addition to A549 cells, a focus was made on several glioma lines.

Comments #7: Variability in plaque assay: Plaque size seem to be quite variable with large standard deviations (Figure 3C). It would be beneficial to clarify how plaques were measured (manual counting, ImageJ, Cell Profiler, etc.).

Authors’ Response #7: the reason for significant variations in the values of plaque sizes within and between the experiments are discussed above. In the Materials and Methods section “4.7. Plaque assay”, it was stated that the plaque size was quantified with Fiji ImageJ. We added the following: “Due to significant variations in the size and morphology of plaques, the plaque contours were manually traced using the tool “freehand selections”, while the plaque areas (µm2) were calculated automatically after setting the size of the scale bar of images as a reference for the actual size (Set scale menu in the Analyze tab).”

Comments #8: Reporting of experimental conditions are inconsistent and sometimes incomplete. It would be beneficial to ensure that timepoints, MOIs, seeding densities are consistently stated across all figures.

Authors’ Response #8: Originally, all relevant information to reproduce the results presented in the manuscript, including time points, MOIs, and seeding densities, were collectively provided for each cell line in the Materials and Methods section and in the corresponding figure graphs and/or figure captions, with the exception of the MOIs for plaque assays, as these must be determined empirically to ensure that plaques are sufficiently distant from each other and do not merge. In the revised version, we suggested a dilution range. We also added the number of cells infected for Western blot analysis in the Materials and Methods section to confirm p19FLAG production, but this information is not critical to reproduce the results. Finally, we added the time points, MOIs, and seeding densities to all figure captions in the revised version of the manuscript.

Comments #9: Depending on the GC3% content of the transgenes, the authors switched between the use of 2 different splice acceptors (IIIaSA and 40SA). However, the rational as to why one splice acceptor would work better than the other for a specific transgene is missing.

Authors’ Response #9: Thank you for pointing this out. The GC3% content of the coding sequence affects mRNA stability and translation efficiency (DOI: 10.15252/embr.201948220). Lower GC3% content is associated with lower translation efficiency, and vice versa. Since it was shown that insertion of transgene coding sequences with 40SA induced their higher production levels than when IIIaSA was used, 40SA was proposed as a tool to enhance the production of transgenes with relatively low GC3% content (doi:10.3390/ijms21145158). We added this discussion to the manuscript.

Comments #10: The discussion section of the article seems to repeat the results section. It would be beneficial to condense the repetitive parts in the discussion.

Authors’ Response #10: In the Discussion section, observations made for each transgene in different assays are compared with previously accumulated data on the same transgenes in the literature. This is where the feeling of duplication of results comes from. However, this is the essence of the Discussion section: to indicate what is new, what is independently confirmed, where there are significant differences in observations and possible causes. Therefore, the team of authors does not see any need to modify the discussion.

Comments #11: There seems to be a lack of consistency in regards to the graphical elements. While some figures include bars and numerical indicators of mean fold change, others do not. Some figures indicate the fold change is non-significant (NS) and other figures simply have no indication for this.

Authors’ Response #11: Fold changes (ratios of means) are designated on graphs where groups are compared to control (parental virus) and/or when this facilitates the perception of differences between groups, particularly for logarithmic scale graphs. In addition, in each figure caption, a threshold value of fold change designated on the graph is stated (e.g., only ≥2-fold differences in the mean IC50 values or only ≥1.3-fold differences in the mean plaque sizes are indicated). Statistics for plaque comparison groups have been excluded from the manuscript.

Comments #12: The method used for the detection of transgene protein is inconsistent. For some transgenes, ELISA assays were used. For others, flow cytometry is used. For p19 protein expression quantification, only a western blot was conducted. Additionally, this western blot is only present in the supplementary data when it should be included the in the manuscript.

Authors’ Response #12: We used the available reagent kits and methods in the lab to compare differences in transgene production or activity. The diversity of methods does not challenge the main conclusion of the section that “The Insertion of Transgenes Downstream of the L3-23K Region Ensures Their Production/Activity at Considerably Higher Levels”. Moreover, the authors who proposed an explanation for the differential impact of transgenes on virus fitness depending on their GC3% content (doi:10.3390/ijms21145158, published in the International Journal of Molecular Sciences) also compared several transgenes and used different methods to confirm their production/activity. Finally, Western blot analysis was performed only to confirm p19Flag production from both the L3-23K and L5-Fib regions without attempting to quantify. If we remove everything that concerns the viruses Ad5Δ24RGD_40SA-p19FLAG (L3-23K) and Ad5Δ24RGD_40SA-p19FLAG (L5-Fiber) from the section, the main conclusion of the section will not change in any way. Hence, this material is additional in meaning, not critical for the main conclusion of the section of the manuscript and can be placed in the Supplemental data file.

Reviewer 2 Report

Comments and Suggestions for Authors

This manuscript performed a comparative analysis of the clinically advanced oncolytic adenovirus Ad5-delta-24-RGD armed with the reporter and therapeutic transgenes using 40SA or IIIaSA with insertion sites downstream of the L3-23K or L5-Fiber region. Of noted, this manuscript provided evidence that the intratumoral production of the human/mouse PD-1 ectodomain alone or in combination with other therapeutic transgenes may promote the generation of an antitumor immune response and could be a promising strategy for enhancing therapeutic efficacy. Collectively, this article should be of great interest to readers of Int. J. Mol. Sci. and I would recommend the publication of this article after addressing minor corrections. 

minor comments/suggestions

  1. Page 2, line 45.Except the oncolytic adenoviruses, oncolytic peptides (Oncolytic immunotherapy) could also activate the immune system, significantly enhancing the anticancer effect of cytotoxic lymphocytes. Over the last two decades, oncolytic peptides have emerged as striking agents to combat cancers. Multiple oncolytic peptides have entered clinical trials for cancer therapy including drug-resistant refractory malignancies. To help readers and potential users, it would be appropriate to cite the representative work on the development of oncolytic peptides (suggest, J. Med. Chem., 2024, 67, 3885. Acta Pharmacol. Sin., 2023, 44, 201. Bioorg. Chem., 2023, 138, 106674.).
  2.  Page 8, line 262.Please change “IC50” to “IC50”.
  3. Page 12, line 445, “The interaction between PD-1 and PD-L1 results in the inhibition of T-cell effector functions such as cytotoxicity, cytokine release, proliferation, and survival”.Blockade of the protein–protein interaction between PD-1 and its ligand PD-L1 has emerged as a promising immunotherapy for anticancer treatments. To help readers and potential users, it would be appropriate to cite the representative work on the development of checkpoint blocking peptides (suggest, Chem, 2024, 10, 2390.).
  4. Due to the potential immunogenicity of mouse proteins in the human body, please discuss the potential immunogenicity of mouse PD-1 ectodomain.
  5. The extracellular domain of PD-1 contains disulfide bonds, which are essential for protein function. How to ensure the correct folding of disulfide bonds in PD-1 ectodomain?
  6. Multiple references lack page numbers (eg, references 35 and 38). Please revise the references according to the standard format.
  7. As the characterization of Ad5-delta-24-RGD expressing is not sufficient to verify the anticancer potential of oncolytic adenovirus, further in vivo experiments may be needed in the future study. The authors should discuss this issue for readers. 

Author Response

Comments #1: This manuscript performed a comparative analysis of the clinically advanced oncolytic adenovirus Ad5-delta-24-RGD armed with the reporter and therapeutic transgenes using 40SA or IIIaSA with insertion sites downstream of the L3-23K or L5-Fiber region. Of noted, this manuscript provided evidence that the intratumoral production of the human/mouse PD-1 ectodomain alone or in combination with other therapeutic transgenes may promote the generation of an antitumor immune response and could be a promising strategy for enhancing therapeutic efficacy. Collectively, this article should be of great interest to readers of Int. J. Mol. Sci. and I would recommend the publication of this article after addressing minor corrections. 

Authors’ Response #1: We sincerely thank the reviewer for taking the time to critically review our manuscript in order to improve the quality of presentation of the scientific material.

Comments #2: Page 2, line 45. Except the oncolytic adenoviruses, oncolytic peptides (Oncolytic immunotherapy) could also activate the immune system, significantly enhancing the anticancer effect of cytotoxic lymphocytes. Over the last two decades, oncolytic peptides have emerged as striking agents to combat cancers. Multiple oncolytic peptides have entered clinical trials for cancer therapy including drug-resistant refractory malignancies. To help readers and potential users, it would be appropriate to cite the representative work on the development of oncolytic peptides (suggest, J. Med. Chem., 2024, 67, 3885. Acta Pharmacol. Sin., 2023, 44, 201. Bioorg. Chem., 2023, 138, 106674.).

Authors’ Response #2: We thank the reviewer for the suggested references. However, oncolytic peptides have no relation to oncolytic adenoviruses and are not the subject of our study. We consider this literature irrelevant to the content of our manuscript.

Comments #3: Page 8, line 262.Please change “IC50” to “IC50”.

Authors’ Response #3: We changed “IC50” to “IC50” in many places in the manuscript. A total of 12 substitutions were made. Thanks for the note!

Comments #4: Page 12, line 445, “The interaction between PD-1 and PD-L1 results in the inhibition of T-cell effector functions such as cytotoxicity, cytokine release, proliferation, and survival”.Blockade of the protein–protein interaction between PD-1 and its ligand PD-L1 has emerged as a promising immunotherapy for anticancer treatments. To help readers and potential users, it would be appropriate to cite the representative work on the development of checkpoint blocking peptides (suggest, Chem, 2024, 10, 2390.).

Authors’ Response #4: We have added this recent comprehensive review article (DOI: 10.3390/jpm14010068) discussing peptides as inhibitors of immune checkpoints.

Comments #5: Due to the potential immunogenicity of mouse proteins in the human body, please discuss the potential immunogenicity of mouse PD-1 ectodomain.

Authors’ Response #5: In our manuscript, we do not propose to the scientific community to use an oncolytic adenovirus producing the mouse PD-1 ectodomain for therapy of human tumors in the clinic. For these purposes, we have constructed oncolytic adenoviruses producing the wild-type human PD-1 ectodomain or mutant human PD-1 ectodomain with high affinity to PD-L1. Moreover, our further studies also do not intend to involve the use of mouse transgenes in the human body. In this regard, the discussion of “the potential immunogenicity of mouse proteins in the human body” is not appropriate in our manuscript.

Comments #6: The extracellular domain of PD-1 contains disulfide bonds, which are essential for protein function. How to ensure the correct folding of disulfide bonds in PD-1 ectodomain?

Authors’ Response #6: According to our data presented in Figure 4B, as well as many previous studies by other groups (doi:10.1073/pnas.1519623112; doi:10.1158/0008-5472.CAN-16-1638; doi:10.1038/mt.2012.252; doi:10.1016/j.ymthe.2019.07.019; doi:10.3389/fimmu.2020.587460), the mouse or human PD-1 ectodomain encoded by the virus has expected biological activity. Therefore, the folding of the PD-1 ectodomain and its secretion in infected cells are normal, and no additional measures are required to ensure the correct folding of disulfide bonds in the PD-1 ectodomain in the case of encoding this transgene in the viral genome. We have never purified a recombinant PD-1 ectodomain and are not aware about the possible existence of a problem with the correct folding of disulfide bonds in the PD-1 ectodomain.

Comments #7: Multiple references lack page numbers (eg, references 35 and 38). Please revise the references according to the standard format.

Authors’ Response #7: Thank you for pointing this out. We have revised the references.

Comments #8: As the characterization of Ad5-delta-24-RGD expressing is not sufficient to verify the anticancer potential of oncolytic adenovirus, further in vivo experiments may be needed in the future study. The authors should discuss this issue for readers. 

Authors’ Response #8: As we have already noted in the discussion section, according to the literature, the intratumoral production of the human or mouse PD-1 ectodomain alone or in combination with other therapeutic transgenes may promote the generation of an antitumor immune response in murine tumor models and could be a viable strategy for enhancing therapeutic efficacy. Formally, the therapeutic efficacy of Ad5-delta-24-RGD, which produces the PD-1 ectodomain, was not tested in vivo in our manuscript, since it is not the direct subject of the manuscript study. In this regard, in the revised version we emphasize that further in vivo experiments are needed to confirm the efficacy of Ad5Δ24RGD_IIIa-IgGκ-PD1 (L3-23K) in murine models, including GL261 and CT-2A gliomas, since Ad5-delta-24-RGD is a clinically advanced virus specifically for the therapy of brain tumors.